# 1*H*-Pyrazolo[3,4-*b*]pyridines: Synthesis and Biomedical Applications

**DOI:** 10.3390/molecules27072237

**Published:** 2022-03-30

**Authors:** Ana Donaire-Arias, Ana Maria Montagut, Raimon Puig de la Bellacasa, Roger Estrada-Tejedor, Jordi Teixidó, José I. Borrell

**Affiliations:** Grup de Química Farmacèutica, IQS School of Engineering, Universitat Ramon Llull, Via Augusta 390, E-08017 Barcelona, Spain; anaariasd@iqs.url.edu (A.D.-A.); ana.montagut@iqs.url.edu (A.M.M.); raimon.puig@iqs.url.edu (R.P.d.l.B.); roger.estrada@iqs.url.edu (R.E.-T.); jordi.teixido@iqs.edu (J.T.)

**Keywords:** 1*H*-pyrazolo[3,4-*b*]pyridines, substitution pattern, synthesis, biological activity

## Abstract

Pyrazolo[3,4-*b*]pyridines are a group of heterocyclic compounds presenting two possible tautomeric forms: the 1*H*- and 2*H*-isomers. More than 300,000 1*H*-pyrazolo[3,4-*b*]pyridines have been described which are included in more than 5500 references (2400 patents) up to date. This review will cover the analysis of the diversity of the substituents present at positions N1, C3, C4, C5, and C6, the synthetic methods used for their synthesis, starting from both a preformed pyrazole or pyridine, and the biomedical applications of such compounds.

## 1. Introduction

Pyrazolo[3,4-*b*]pyridines are one of the bicyclic heterocyclic compounds which are members of the family of pyrazolopyridines formed by five congeners (the [3,4-*b*], [3,4-*c*], [4,3-*c*], [4,3-*b*], and [1,5-*a*]), which are the possible fusions of a pyrazole and a pyridine ring [1]. They can present two isomeric structures: 1*H*-pyrazolo[3,4-*b*]pyridines (**1**) and 2*H*-pyrazolo[3,4-*b*]pyridines (**2**) (Figure 1). The first monosubstituted 1*H*-pyrazolo[3,4-*b*]pyridine (R^3^ = Ph) was synthesized by Ortoleva in 1908 upon treatment of diphenylhydrazone and pyridine with iodine [2]. Only three years later, Bulow synthesized three *N*-phenyl-3-methyl substituted derivatives **1** (R^1^ = Ph, R^3^ = Me), starting from 1-phenyl-3-methyl-5-amino-pyrazole which was treated with 1,3-diketones in glacial AcOH, following a widely used strategy [3].

Since then, structures **1** and **2** have attracted the interest of medicinal chemists due to the close similitude with the purine bases adenine and guanine, an interest that is clearly illustrated by the more than 300,000 structures **1** (5000 references, including nearly 2400 patents) and around 83,000 structures **2** (nearly 2700 references, including 1500 patents) included in SciFinder [4] and 180 reviews, including some on the synthesis of such structures [5,6,7,8]. These compounds present up to five diversity centers that allow a wide range of possible combinations of substituents capable of addressing different biological activities.

Our group, like others, was attracted by the versatility of such compounds and used them as scaffolds for the synthesis of tyrosine kinase inhibitors (TKI). The analysis of the different reviews accessible in the literature showed that, to the best of our knowledge, there is no review covering both the synthetic methods used for their synthesis and the biological activities achieved with these structures.

Consequently, we decided to carry out a review of the literature covering the structural, synthetic, and biological aspects of pyrazolo[3,4-*b*]pyridines, but previously, we analyzed the information available, taking some preliminary decisions:(a)The review will cover only those compounds that present a fully unsaturated pyridine ring, not taking into account other degrees of unsaturation. Nevertheless, the presence of hydroxy groups at C4 or C6 will be covered, and consequently, the corresponding pyridone tautomers will be included due to the greater stability of the 4-oxo or 6-oxo derivatives.(b)In the case of pyrazolo[3,4-*b*]pyridines not substituted at the nitrogen atoms of the pyrazole ring, two tautomeric forms are possible: the 1*H*- (**1**, R^1^ = H) and the 2*H*-pyrazolo[3,4-*b*]pyridine (**2**, R^2^ = H) (Figure 2). Although such tautomerism could complicate the diversity analysis of such compounds, the AM1 calculations of Alkorta and Elguero clearly showed the greater stability of the 1*H*-tautomer by a difference of 37.03 kJ/mol (almost 9 kcal/mol) [9].

Although the number of pyrazolopyridines **2** seems to be around 83,000, most of them are compounds with R^2^ = H. When the search is repeated with the elimination of tautomerism, only about 4900 2*H*-pyrazolo[3,4-*b*]pyridines **2** (R^2^ = H) remain, included in only 130 references. In many cases, the 2-NH tautomer is only favored when the pyridine ring is not fully dehydrogenated, thus being a tetrahydropyridone [10,11]. Such behavior is in agreement with our experimental results and DFT calculations for C4–C5 fused pyrazole-3-amines, which pointed out that the only way to majorly observe the 2*H*-tautomer is to have the pyrazole ring fused to a non-aromatic ring [12]. Thus, the N1 substituted isomers **1** present aromatic circulation in both rings, thanks to the double bond that can be drawn in the fusion of both rings while N2 substituted structures **2** only allow a peripheric circulation due to the positions of the double bonds in the pyrazole ring. Therefore, the different aromatic circulation has a high impact on the relative stability of the isomers.

Convergently, the total number of the 2-substituted derivatives **2** is drastically reduced to around 19,000, which includes R^2^ = Me (18.28%, 110 references) [13], R^2^ = Ph (8.58%, 213 references) [14], and R^2^ = heterocycle (56.86%, 35 references) [15]. In most cases, the regiospecificity is achieved by starting from an adequately substituted pyrazole.

Moreover, a search carried out at DrugBank (https://www.drugbank.ca/, accessed on 8 March 2022), a database containing information on drugs and drug targets, has revealed that there are 14 1*H*-pyrazolo[3,4-*b*]pyridines **1** in different phases of research: 7 are classified as Experimental (having shown biological activity), 5 are classified as Investigational (included in some of the approval phases), and 2 have already been approved. On the contrary, there is not a single 2*H*-pyrazolo[3,4-*b*]pyridine **2** in any of these stages.

(c)To the best of our knowledge, there are only five specific reviews on pyrazolopyridines prior to this review. Three of them are devoted to the synthesis of such compounds [6,7,8], the most recent being from 2012 [6]. The other two cover biological aspects, either from a general perspective (a review from 1985, [16]) or a very specific point of view (kinase inhibitors, 2013 [17]). Furthermore, 5591 references cover the 300,000 1*H*-pyrazolo[3,4-*b*]pyridines included in SciFinder, but 3005 of them (almost 54%) are from 2012 or later (1413 being patents).

Consequently, taking into account the preceding considerations, the rather low number of 2*H*-pyrazolo[3,4-*b*]pyridines **2**, and the low impact of such isomers in drugs under development, we decided to focus the present review only on the structural, synthetic, and biological aspects of 1*H*-pyrazolo[3,4-*b*]pyridines **1**, paying special attention to the period from 2012 to 2022.

## 2. Structural Features of 1*H*-Pyrazolo[3,4-*b*]pyridines: Substitution Patterns

The first aspect addressed in this review is the analysis of the diversity already covered at the different substitution positions present in the Markush formula of **1** by the molecules included in SciFinder. Such information is not directly accessible from SciFinder or other computerized databases due to the huge number of structures **1** included (more than 300,000), so analysis with specialized software is not possible. Consequently, we have explored the substitution patterns at N1, C3, C4, C5, and C6 one by one in order to obtain a picture of the diversity covered.

### 2.1. Substitution Pattern at N1

The analysis of the diversity at N1 included in Table 1 shows that almost one-third of the more than 300,000 compounds **1** have a methyl group at N1 followed by those presenting other alkyl groups (around 23%) or a phenyl group (15%). The number of unsubstituted pyrazoles is around 20%. Such distribution agrees with the synthetic methods used for the synthesis of compounds **1** from a pyrazole ring that requires to be substituted to avoid the formation of two regioisomers (see Section 3).

### 2.2. Substitution Pattern at C3

The diversity analysis included in Table 2 clearly shows that more than three-quarters of the diversity is covered by the substituents hydrogen and methyl. Other substituents, such as phenyl, cycloalkyl, and heterocyclic rings, on one side, or an amino group and a hydroxy group as the carbonyl tautomer on the other side, only have minor contributions. Such a ratio is directly connected with the more employed synthetic methodologies used for the construction of these heterocycles although the importance of amino and carbonyl groups in the biological activities of pyrazolopyridines cannot be dismissed.

**Table 2 molecules-27-02237-t002:** Substitution pattern at C3 of 1*H*-pyrazolo[3,4-*b*]pyridines **1**.

R^3^	Structures 1 (%)	Number of References	Selected References
H	30.83	2049	[19,28]
Me	46.77	1395	[29,30]
Cycloalkyl	4.01	98	[31,32]
Ph	2.12	540	[27,33]
Heterocycle	4.88	1138	[32,33]
NH_2_	4.69	727	[18,34]
OH	2.17	239	[35,36]
Other	4.52	-	-

### 2.3. Substitution Pattern at C4, C5, and C6

The chemical diversity present at C4, C5, and C6 (Table 3, Table 4 and Table 5) is usually interconnected because it depends on the building block used for the construction of the pyridine ring that usually is an α,β-unsaturated ketone, or a 1,3-dicarbonyl compound. The percentages of the R^4^ substituents are in agreement with such a synthetic approach, that being mainly a hydrogen atom, methyl, phenyl, or heterocyclic ring. The hydroxy group as the keto tautomer shows the use of ester groups during the formation of the pyridine ring that finally ends as a pyridone. A different origin has the amido group present at C4 that usually comes from the manipulation of a cyano or ester group present at such a position and not from the direct formation of the pyridine ring.

**Table 3 molecules-27-02237-t003:** Substitution pattern at C4 of 1*H*-pyrazolo[3,4-*b*]pyridines **1**.

R^4^	Structures 1 (%)	Number of References	Selected References
H	37.33	3003	[37,38]
Me	6.59	389	[18,27]
Ph	2.29	660	[10,39]
Heterocycle	2.42	355	[27,40]
OH	0.89	312	[41,42]
N-substituent	4.54	980	[43,44]
CONHR	38.30	148	[29,45]
Other	7.63	-	-

In the case of C5, the most usual substituent is a hydrogen atom because, in most of the cases, the starting α,β-unsaturated ketone, or a 1,3-dicarbonyl compound has no substituent at α-position of the corresponding compound. Only in a small percentage of cases is there a substituent at such a position. Once more, the presence of amide groups at C5 is remarkably high, a group that usually comes from the manipulation of a cyano group conveniently introduced at such a position.

**Table 4 molecules-27-02237-t004:** Substitution pattern at C5 of 1*H*-pyrazolo[3,4-*b*]pyridines **1**.

R^5^	Structures 1 (%)	Number of References	Selected References
H	58.61	2753	[26,34]
Me	0.84	87	[46,47]
Ph	0.66	211	[19,48]
Heterocycle	3.18	349	[49,50]
N-substituent	4.90	231	[51,52]
Halogen	1.34	583	[18,19]
CONHR	12.13	430	[53,54]
Other	18.34	-	-

Finally, as for the diversity at C6, the distribution of substituents agrees with the main use of α,β-unsaturated ketones where the end substituent of the system is a hydrogen atom, a methyl, phenyl, or heterocyclic ring.

**Table 5 molecules-27-02237-t005:** Substitution pattern at C6 of 1*H*-pyrazolo[3,4-*b*]pyridines **1**.

R^6^	Structures 1 (%)	Number of References	Selected References
H	35.10	3270	[19,55]
Me	20.15	679	[18,27]
Ph	12.44	509	[33,56]
Heterocycle	8.65	300	[57,58]
OH	1.52	334	[59,60]
N-substituent	3.22	565	[61,62]
Carbonyl group	1.27	92	[63,64]
Other	17.64	-	-

Once the diversity of the substituents at the various positions of the 1*H*-pyrazolo[3,4-*b*]pyridines has been analyzed, a more visual comparison of the diversities covered at positions N1, C3, C4, C5, and C6 of structures **1** included in SciFinder is possible, as shown in Figure 3.

Although the one-by-one analysis of the substituents present at the different positions of the pyrazolopyridine system gives a good picture of the diversity already covered, it would certainly be more interesting to have an idea of the more common di- and trisubstitution patterns. Therefore, a search of the most common combinations of substituents at C3, C4, and C5 was carried out with the following results: 46.83% of the compounds described correspond to a 4,6-disubstituted pattern, 22.04% to 5-monosubstituted compounds, 7.76% to 3,4,5-trisubstituted structures, and 3.37% to unsubstituted pyrazolopyridines.

To see the correlation between such preferred substitution patterns at the pyridine ring and the corresponding substituents at N1 and C3, we have prepared Figure 4 and Figure 5, which correspond to the most abundant 4,6-disubstituted and 5-monosubstituted patterns.

In the case of the 4,6-disubstituted 1*H*-pyrazolo[3,4-*b*]pyridines, the most common substituent at position C3 is a methyl group (66.06%) followed by a hydrogen atom (23.69%). Correspondingly, for R^3^ = H, the most abundant R^1^ is a methyl group (14.07%) and an alkyl group (64.66%), while for R^3^ = Me the most abundant R^1^ is a methyl group (43.20%), an alkyl group (23.89%), or a phenyl group (25.74%). These results correlate with the use of a 1-substituted 3-methyl-1*H*-pyrazole or a 1-substituted 1*H*-pyrazole as starting material to afford the 3-methyl (R^3^ = Me) and 3-unsubstituted (R^3^ = H) 4,6-disubstituted 1*H*-pyrazolo[3,4-*b*]pyridines **1**, respectively.

On the other hand, the situation with the substituents being more abundant at position C3 of the 5-monosubstituted 1*H*-pyrazolo[3,4-*b*]pyridines is more equilibrated, as the hydrogen atom (35.20%) and methyl group (37.20%) are virtually tied. Moreover, the presence of R^1^ = H reaches higher values than in the case of the 4,6-disubstituted pyrazolopyridines: 21.82% when R^3^ = H and 19.47% when R^3^ = Me. Once more, the preferred substituents at R^1^ are the methyl group (55.37%) when R^3^ = Me and the alkyl group (44.96%) when R^3^ = H.

## 3. Synthetic Approaches to 1*H*-Pyrazolo[3,4-*b*]pyridines

1*H*-Pyrazolo[3,4-*b*]pyridines **1** are bicyclic heterocyclic systems, and therefore, there are many different strategies to achieve such a structure. This review, however, will focus on two major strategies: the formation of a pyridine ring into an existing pyrazole ring **3** and the formation of the pyrazole ring into a preexisting pyridine ring **4** (Figure 1).

Therefore, it is possible to make a first classification of the synthetic methods depending on which of the two strategies shown in Figure 1 is being used.

### 3.1. Pyridine Formation onto a Preexisting Pyrazole Ring

Those reactions are characterized by using different pyrazole derivatives to synthesize the pyridine ring. Most of the reactions that can be found in the literature use 3-aminopyrazole as the starting material, which generally acts as a 1,3-NCC-dinucleophile, reacting therefore with a 1,3-CCC-biselectrophile. The different reactions will be classified according to the nature of the 1,3-CCC-biselectrophile used.

#### 3.1.1. 1,3-Dicarbonyl Compounds and Derivatives as 1,3-CCC-Biselectrophiles

Dicarbonyl compounds **5** have two electrophilic positions (the carbonyl groups) and a nucleophilic one (the α position in between the two carbonylic groups). Since the compound reacts as a 1,3-CCC-biselectrophile, the two carbonyl groups are the ones involved in the reaction. There is a strong debate on the mechanism for this type of reaction and two different proposals arise from literature as commented below (Figure 2).

5-Aminopyrazole **6** has two different reactive points that can act as a nucleophile: the amino group (-NH_2_) and the sp^2^ carbon at its β position. The reaction starts with the nucleophilic attack of one of those nucleophiles onto one of the two carbonyl groups followed by dehydration. Unfortunately, there is no consensus regarding which of the two nucleophiles reacts in the first place. The first attack is followed by the second nucleophile reacting with the unreacted carbonyl group, leading to the formation of a 6-member ring **7a** or **7b** (depending on the initial attack) which, after dehydration, forms the pyrazolo[3,4-*b*]pyridine system **1**. This mechanism is the one taking place for the majority of dicarbonyl compounds and derivatives, adapting it to the nature of each reactant.

The vast majority of the reactions reported in literature follow the same conditions (Figure 3). AcOH is commonly used as a solvent, and the reaction is carried out at reflux temperature or microwave (MW) irradiation, with reaction times that may vary depending on the substituents present [65,66]. It is also possible to carry out this reaction using water as a solvent at 90 °C for 16 h [67] or using MeOH and HCl at room temperature for 16 h [68].

As it can be seen from the reaction mechanism, if the 1,3-dicarbonyl compound is nonsymmetrical, two regioisomers can be formed [66]. The proportions among the products will depend on the relative electrophilicity of the two carbonyl groups. If the two are very similar, the proportions are going to be near 50%, but if they are very different, it is possible to carry out the reaction having regioselectivity higher than 80%.

As mentioned, there is no consensus on the order of the attack of the two nucleophiles, and it is even possible to find authors performing very similar reactions but claiming opposite results, obtaining yields higher than 60% in both cases. On the one hand, Ghaedi et al. performed the reaction using ethyl 2,4-dioxo-4-phenylbutanoate derivatives as starting materials, achieving final molecules with yields ranging from 60 to 90%. The resulting 1*H*-pyrazolo[3,4-*b*]pyridines present a COOEt group at the R^6^ position [63]. On the other hand, Ibrahim et al. used ethyl acetoacetate derivatives as starting materials, compounds that present one carbonyl group with higher electrophilicity than the other, as it so happens, with 2,4-dioxo-4-phenylbutanoate derivatives. The reaction was carried out using the same conditions above, but the results ended up being the opposite. The molecules described by Ibrahim, with yields ranging from 62 to 76%, seem to have the hydroxy group at the R^4^ position, and not at the R^6^ as described by Ghaedi [42]. This example shows how difficult it can be to determine not only the mechanism, but also the structure of the molecules obtained. Since there are no big differences between the NMR spectra of the two regioisomers, it might be difficult to determine which one is present unless both are available.

To clarify the mechanism and avoid the presence of both regioisomers, Emelina et al. used 1,1,1-trifluoropentane-2,4-dione (**8**) and its derivatives to differentiate the two carbonyl groups present in the initial reactant (Figure 4) [66].

They concluded that, since the carbonyl group having the CF_3_ group is more electrophilic, it should be the one reacting in the first place. The results show that, once the pyridine ring gets formed, the CF_3_ stays at the R^4^ position, meaning that the amino group from the 5-aminopyrazole **6** reacts in second place. To assign the position of the CF_3_ group, they used ^1^H-NMR and ^13^C-NMR to look after the long-range coupling constants between the fluorine atoms from the CF_3_ group and the H and C atoms from the R^3^ group [66].

This type of reaction can be performed with a wide range of 1,3-CCC-biselectrophiles, such as the ones included in Table 6.

All of the 1,3-CCC-biselectrophiles listed in Table 6 are open-chain compounds; however, some six-membered cyclic 1,3-CCC-biselectrophiles that get opened during the reaction have also been used to construct the 1*H*-pyrazolo[3,4-*b*]pyridine skeleton (Figure 6).

In the case of 2,3-dihydro-4*H*-pyran-4-ones **8**, the reaction works in a very similar way as the other 1,3-dicarbonyl compounds, but with 4-hydroxy-2*H*-pyran-2-ones **9**, the reaction involves a decarboxylation, as depicted in Figure 5.

The conditions for those reactions vary depending on the author, Jouha et al., that used molecule **9** as the initial reactant, carrying out the reaction under microwave irradiation at 180 °C for 3 h using BuOH the solvent and TsOH as the catalyst. The final product presented methyl groups at positions R^3^, R^4^, and R^6^; therefore, no regioselectivity issues were present, and the final product was obtained with a 98% yield [69]. Ianoshenko et al. published three different papers about this matter, proposing two different reaction conditions. On one hand, they tested the same conditions used for the 1,3-dicarbonyl reactants, using AcOH at reflux temperature for 1 h, obtaining yields up to 98%. On the other hand, they used DMF, TMSCl at 100 °C for 1 h, obtaining yields up to 93% [70,71,72].

In none of the cases were regioselectivity problems reported; this could be due to either a mistake in analyzing the results or the higher selectivity of this subtype of reactions over the traditional 1,3-dicarbonyl condensations described above.

#### 3.1.2. Michael Acceptors Used as 1,3-CCC-Biselectrophiles

α,β-Unsaturated ketones **10** have also been used as 1,3-CCC-biselectrophiles in the formation of 1*H*-pyrazolo[3,4-*b*]pyridines **1** by reacting them with 5-aminopyrazole **6**. Michael acceptors react similarly to 1,3-dicarbonyl compounds (Figure 6).

The sp^2^ carbon in β to the amino group is the one believed to be the most nucleophile and, therefore, is the one that seems to attack in the first place, performing a Michael addition. Unfortunately, there is no complete agreement on that matter. Accordingly, the amino group (NH_2_) would react in the second place, attacking the carbonyl group of the Michael acceptor, and leaving a hydroxyl group. After the elimination of water and spontaneous oxidation, the pyrazolo[3,4-*b*]pyridine would be formed. This last spontaneous oxidation step takes place in more than one of the mechanisms proposed in the literature. It is not clear how the oxidation proceeds, but some hypotheses have been made. Among them, the most plausible is oxidation due to the atmospheric oxygen. Alternatively, some authors propose a disproportion of the molecule, but in this case, a 50:50 mixture of the oxidized and reduced products would be obtained, a result not clearly described in any of the manuscripts.

Such a reaction is carried out under both acidic and basic conditions. Stepaniuk et al. compared different reaction conditions, selecting either acetic acid at reflux for 12 h or HCl/1,4-dioxane in EtOH at 100 °C for 18 h, with yields ranging from 44 to 99% [73]. Han et al. performed the reaction using 1.0 M NaOH in glycol as a solvent at 120 °C for 5–12 min, affording in all cases yields above 90% [74]. Many authors have used Lewis acids as catalysts, CuCl_2_, ZrCl_4,_ or ZnCl_2_ [75,76,77], instead of Brønsted–Lowry acids. Shi et al. performed this reaction without a catalyst by using the ionic liquid [bmim]Br as a solvent, keeping the yields between 80 and 96% [78].

Since this reaction is the result of the attack of two nucleophilic centers on two electrophilic centers, it is also possible to achieve two different regioisomers. Stepaniuk et al. performed an analysis of the effect of the reaction conditions on the final regioisomer ratios. They concluded that the regioselectivity of the reaction was very sensitive to small changes, not only on the conditions, but also on the structure of the initial reactants [73].

#### 3.1.3. Diethyl 2-(Ethoxymethylene)malonate as 1,3-CCC-Biselectrophile (Gould–Jacobs Reaction)

The Gould–Jacobs reaction is often used for the synthesis of quinolines or 4-hydroxyquinoline derivatives using aniline and diethyl 2-(ethoxymethylene)malonate **11** as starting materials. 1*H*-Pyrazolo[3,4-*b*]pyridines can be obtained by using 3-aminopyrazole **6** (or its derivatives) instead of aniline. The product obtained, in the majority of the cases, is a 4-chloro substituted 1*H*-pyrazolo[3,4-*b*]pyridine **1** [79,80,81,82,83].

The mechanism proposed in the literature for the formation of 1*H*-pyrazolo[3,4-*b*]pyridines **1** by the Gould–Jacobs reaction is depicted in Figure 7.

In this case, the amino group of the 3-aminopyrazole **6** would react in the first place, attacking the enol ether group present in **11,** causing the elimination of ethanol. The subsequent nucleophilic attack on one of the two ester groups present eliminates ethanol as well. Finally, the carbonyl group present in the 6-membered ring reacts with POCl_3_, forming the corresponding 4-chloro-1*H*-pyrazolo[3,4-*b*]pyridines **1**. Since the 1,3-CCC-biselectrophile is symmetrical, there are no regioselectivity problems.

The reaction conditions used for such protocols are often very similar. The reaction can be performed by using ethanol as solvent at reflux temperature, followed by the treatment with POCl_3_ [79,82]. Since diethyl 2-(ethoxymethylene)malonate **11** is liquid at room temperature, it is possible to perform the reaction without any solvent, using 100–110 °C for times between 1.5 and 12 h [80,81,83,84]. Rimland et al. performed the reaction without any solvent at 160 °C for 5 h using SOCl_2_ instead of POCl_3_ in the last step [83].

As the electrophile used for the reaction is always the same, the variations in the structure of the final product come from the substituents present in the pyrazole reagent (R^1^ and R^3^). In all cases, the final product is 4-chloro substituted 1*H*-pyrazolo[3,4-*b*]pyridine **1**, except for Pan et al., which obtained the 4-hydroxy substituted compound because the POCl_3_ was not included [84].

The Gould–Jacobs reaction is a simple way to achieve 4-chloro-1*H*-pyrazolo[3,4-*b*]pyridines, but it is not very versatile due to the limitations in the nature of the substituents accessible. Nevertheless, it is an interesting strategy for further derivatization of the 1*H*-pyrazolo[3,4-*b*]pyridines formed.

#### 3.1.4. In Situ Formation of the 1,3-CCC-Biselectrophiles or the 1,3-NCC-Dinucleophiles

One way to overcome regioselectivity problems is to generate in situ the 1,3-CCC-biselectrophile by using an aldehyde **12**, a carbonyl compound **13** bearing at least an α-hydrogen atom, and a conveniently substituted pyrazole **6**. Many authors have described the use of such a three-component reaction in very high yields without reporting regioselectivity issues, agreeing, in all cases, on how the mechanism is taking place (Figure 8) and which isomer is being formed [40,64,85,86,87,88,89,90].

The reaction starts with the formation of the 1,3-CCC-biselectrophile by a carbonyl condensation between the α-carbon of the carbonyl compound **13** (usually a ketone) and the aldehyde **12,** followed by the elimination of water. Once the electrophile is formed, the mechanism is similar to the one described in Figure 6, including a Michael addition of the 5-aminopyrazole **6** to the in-situ-formed α,β-unsaturated compound, followed by a closing of the pyridine ring with the subsequent elimination of water and final oxidation to yield the 1*H*-pyrazolo[3,4-*b*]pyridine scaffold **1**. Ezzrati et al. performed this reaction in both the presence and absence of air. When the reaction was carried out in the absence of air at 50 °C during 40–60 min, the final molecule contained a dihydropyridine instead of a pyridine ring. They were able to convert such a compound to the desired pyrazolopyridine by dissolving the intermediate in ethanol at reflux for 10–30 min in the presence of air [85]. Such a result leads to the conclusion that air may be necessary to carry out the oxidation reaction.

This protocol is frequently carried out with a catalyst. Shi et al. used *L*-proline to help the carbonyl condensation by forming an imine between the amino group of the *L*-proline and the aldehyde. The reaction is carried out using EtOH as solvent at 80 °C for 30–60 min [89]. Acids or bases are commonly used as catalysts as well—to assist with deprotonation in case of a base, or to increase electrophilicity if an acid is used. El-borai et al. used acetic acid (or a mixture of acetic acid and triethylamine) to catalyze the reaction, using high temperatures (150 to 160 °C). The reaction takes 15 to 20 min to complete, with yields ranging from 65 to 88% when only acetic acid is present, and from 86 to 98% when a combination of both catalysts is used. It is important to mention that the initial reactants were not the same and thus the different yields could be due to other elements besides the present catalysts [64].

To make the reaction greener, some authors have used ionic liquids as both solvents and catalysts [40,87,90]. Jadhav et al. used [Et_3_NH][HSO_4_] as the ionic liquid, demonstrating that it can be recycled up to five times without losing its capability of performing the reaction. This is a very interesting approach in terms of waste reduction, lower temperatures (60 °C), higher yields (90–96%), and safer reaction conditions [40]. The reaction has also been carried out without a catalyst by Ezzrati and Rahmati et al., using ethanol and water as solvents, respectively, or even without any solvent or catalyst by Quiroga et al. under microwave irradiation at 200 °C during 9 min [85,86,88].

There are some examples of the 1,3-NCC-dinucleophile being generated in situ. Hamama et al. and Marzouk et al. used 3-pyrazolones that reacted with ammonia to generate the 3-aminopyrazole ring [91,92].

### 3.2. Pyrazole Formation onto a Preexisting Pyridine Ring

As previously commented, pyrazolo[3,4-*b*]pyridines can be also synthesized starting from a pyridine derivative and closing the pyrazole ring. Those reactions are usually carried out using hydrazine **17** (or substituted hydrazine) and a pyridine ring **14–16** containing a good leaving group at position C2 and an electrophilic group at position C3 (Figure 9).

The leaving group most widely used in these reactions is a chlorine atom, while the electrophilic group is chosen among carbonyl groups (aldehyde, ketone, or ester groups) or a cyano group. Depending on the electrophilic group, the final compounds **18**–**20** present as the R^3^ substituent a hydrogen atom, an alkyl or aryl group, a carbonyl group, or an amino group, respectively. The other diversity centers included in the final pyrazolo[3,4-*b*]pyridine must be present in the initial pyridine (R^4^, R^5^, and R^6^) and the hydrazine (R^1^).

The mechanism is very similar for the three different options. In all the cases, the reaction starts with the nucleophilic attack of the hydrazine **17** to both electrophilic positions of the corresponding pyridine ring **14**–**16**: the chloro substituted carbon and the electrophilic group at position C3. After this step, the corresponding intermediate evolves to the final 1*H*-pyrazolo[3,4-*b*]pyridines **18**–**20**, either by tautomerization (in the case of a C3 nitrile or ester group) or by elimination of water (in the case of aldehydes or ketones). It is important to mention that it is not clear in which tautomeric form is present the carbonyl group at position C3 since some authors draw the keto form and others draw the enol form.

Since the reactions are very similar regardless of the groups present at C3 of the pyridine ring, the reaction conditions are also very similar. As an example, Figure 10 shows the conditions mostly used for the reaction:

Most authors use hydrazine hydrate as the initial reactant, using ethanol as solvent at reflux temperature; the reaction time needed ranges from 2 h to more than 15 h [93,94,95,96]. Arafa and Hussein performed the reaction at 50 °C using a sonicator, allowing them to obtain the product in 15 min [97]. Since hydrazine hydrate is liquid at room temperature, some authors perform the reaction without a solvent, with the reaction times going from 3 to 10 h [98,99,100,101].

There is too much variation among the reaction conditions used to compare the two methods and properly discuss a decrease in the reaction time by eliminating the solvent. Mali et al. performed the reaction without solvent at reflux temperature and under microwave irradiation, thus reducing the reaction time from 7–10 h to 1.5–2 h [101]. Orlikova et al. also performed the reaction without solvent but at 150 °C, and using ethanol as a solvent at reflux temperature; in both cases, the reaction times were very similar: 2 h without solvent and 2–3 h using ethanol.

Other solvents can be used to perform the reaction, like DMF, either at reflux temperature or around 100 °C [102,103]; BuOH at reflux temperature [104]; or ethylene glycol at 165 °C [105,106]. It is also possible to find authors who have used acids or bases to favor the reaction. Thus Al-kaabi and Elgemeie used Et_3_N, and the reaction was complete after 3 h [107], and Teixeira et al. and Selvi et al. used TsOH at reflux (3 h) and microwave-assisted heating (4–13 min, 320 W), respectively [38,108].

As can be seen, there is not a standardized method for carrying out this type of reaction.

Table 7 includes the different pyridine reactants that have been used.

### 3.3. Other Reactions

The above classification has been made to systematize the reactions used to synthesize 1*H*-pyrazolo[3,4-*b*]pyridines **1**. Even though most of the references that can be found in the literature can be classified in that way, there is a pool of reactions that do not fit into any of the categories mentioned [91,109,110,111,112,113,114,115,116].

## 4. Biomedical Applications of 1*H*-Pyrazolo[3,4-*b*]pyridines

1*H*-Pyrazolo[3,4-*b*]pyridines **1** have been extensively used as a scaffold for the synthesis of small molecules looking for therapeutic properties to treat different diseases. From all the molecules containing a pyrazolo[3,4-*b*]pyridine core (more than 300,000 reported molecules), 156,660 molecules have been synthesized for therapeutic purposes.

According to the literature, the most relevant, important part of the overall biomedical applications’ bioactivity indicators for 1*H*-pyrazolo[3,4-*b*]pyridines are antitumor agents (22,675 molecules), anti-inflammatory agents (19,416 molecules), and nervous system agents (14,203 molecules). More concretely, they account for 38% of the biomedical applications (Figure 7).

In the following part of this review, we focus on these three main biomedical applications because they represent more than a third of the overall bioactivity indicators, as well as the different pattern substitutions of the pyrazolo[3,4-*b*]pyridine scaffold selected for each disease, giving more importance to the major ones.

### 4.1. Antitumor Agents

The use of 1*H*-pyrazolo[3,4-*b*]pyridines **1** as a scaffold for antitumor agents corresponds to 15% of the biomedical applications. In most of them, the selected substituents are at R^3^ and R^5^ (8112 molecules, 36% of this group of compounds), followed by disubstitution at R^3^ and R^4^ (1375 molecules), monosubstitution at R^3^ (1208 molecules), and other substitution patterns, which include around 1000 molecules each. An analysis of the type of substituents present in the most abundant group has shown that 5897 structures (almost 26% of the structures reported with anticancer activity) present a combination of substituents in which R^1^ = R^4^ = R^6^ = H, and R^3^ and R^5^ are heterocyclic rings.

1*H*-Pyrazolo[3,4-*b*]pyridines derivatives **21**, **22**, and **23** (Figure 8) have been described as potent inhibitors of hGSK-3α (IC_50_ = 56 ± 6 nM, 18 ± 2 nM, and 11 ± 2 nM, respectively) [94]. Further optimization of these promising small molecules was performed, resulting in molecule **24** with an IC_50_ of 0.8 ± 0.4 nM [117].

Cyclin-dependent kinases (CDKs) are encouraging drug targets for various human diseases, in particular for cancer. SAR studies of compounds **25** and **26** (Figure 9) led to the discovery of an excellent CDK1/CDK2 selective inhibitor **27**, BMS-265246 (CDK1/cycB IC_50_ = 6 nM and CDK2/cycE IC_50_ = 9 nM) [118].

Compound **28** inhibits CDK1 activity with an IC_50_ of 23 nM. This inhibitory activity has been observed by a reduced amount of 33P-cATP incorporated into the immobilized substrate in a FlashPlate assay format. **28** has also shown inhibition growth of HeLa cervical adenocarcinoma, A375 malignant melanoma, and HCT-116 colon carcinoma cells, with IC_50_ values of 1.7, 0.87, and 0.55 µM, respectively. Moreover, this compound also exhibits inhibition of VEGFR-2 kinase, a receptor tyrosine kinase implicated in angiogenesis, another important mechanism for tumor progression with an IC_50_ value of 1.46 µM [119].

In the growth of colorectal cancer, multiple lines of evidence propose that the mediator-complex-associated, cyclin-dependent kinase (CDK8) may act as an oncogene. Compound **29** (MSC2530818) is a structure-based, designed small molecule that exhibits outstanding kinase selectivity, biochemical and cellular potency, microsomal stability, and is orally bioavailable. This compound shows an in-vivo, oral, pharmacokinetic profile in mice, rats, and dogs. It also evidences reduction of the tumor-growth rates of established human SW620 colorectal carcinoma xenografts using two different oral dosing schedules. Considering its huge activity, compound **29** went into preclinical, in-vivo, animal efficacy and safety studies [53].

Molecule **30** (CAN508) has been used for a scaffold-hopping strategy, and the scaffold alteration using the 1*H*-pyrazolo[3,4-*b*]pyridine core appears to cause a positive alteration in the selectivity profile of the inhibitors. Compound **31** exhibits an excellent inhibitor activity against CDK2 and CDK9 (IC_50_ values of 0.36 µM and 1.8 µM, respectively). Furthermore, compound **32** evidences extraordinary selectivity towards CDK2 (265-fold over CDK9) [120].

1*H*-Pyrazolo[3,4-*b*]pyridine analogs **33** and **34** also manifest a selective and potent cyclin-dependent kinase and cellular antiproliferative inhibition. These compounds have shown an excellent in-vitro inhibition of the cellular proliferation in HeLa, HCT116, and A375 human tumor cell lines [121].

About 7% of all cancers present the V600E mutation of B-Raf kinase, which results in the constitutive activation of the MAPK signaling pathway. Molecules **35** and **36** (Figure 10) have been structure-based designed for inhibiting B-Raf^V600E^. These compounds have shown to be potent, selective, and orally bioavailable debutants that inhibited tumor growth in a mouse xenograft model driven by B-Raf^V600E^ [67].

Moreover, one of the most difficult types of cancer to treat is metastatic melanoma; the inhibition of BRAF^V600E^ mutant kinase is its main current therapy. Nevertheless, the inhibition of BRAF by small molecules in cancer patients provokes an increase of wild-type BRAF activity in healthy tissue, causing side effects and the formation of new tumors. Hoorens et al. have developed the BRAF^V600E^ kinase inhibitor **37** (Figure 10), the activity of which can be switched on and off in a reversible way with light. Consequently, the drug can be selectively activated at the desired site of action, avoiding side effects. This small molecule contains in its structure an azobenzene photoswitch that, once activated with light, increases the inhibitory activity by 10-fold compared with the non-activated form [122].

Lead compounds **38** and **39** (Figure 11) have been described for their strong inhibition activity towards a broad spectrum of Bcr-Abl mutants, such as the gatekeeper T315I and p-loop mutations, which are associated with disease progression in chronic myelogenous leukemia (CML). They resolutely inhibited the kinase activities of Bcr-Abl^WT^ and Bcr-Abl^T315I^ with IC_50_ values of 0.60, 0.36, and 1.12, 0.98 nM, respectively [123].

GZD824 **40** is another small molecule described for the treatment of CML that includes a 1*H*-pyrazolo[3,4-*b*]pyridine moiety. Such a compound is an orally bioavailable inhibitor against a broad spectrum of Bcr-Abl mutants, including T315I. This promising compound has a K*_d_* value of 0.32 and 0.71 nM for Bcr-Abl^WT^ and Bcr-AblT315I, respectively. It also effectively suppresses the proliferation of Bcr-Abl-positive K562 and Ku812 human CML cells with IC_50_ values of 0.2 and 0.13 nM, respectively. Its bioavailability is 48.7%, and it presents a half-life of 10.6 h. Moreover, it stimulates tumor regression in mouse xenograft tumor models and greatly improves the survival of mice, altogether making GZD824 a potential lead aspirant for the development of Bcr-Abl inhibitors [95].

A series of 1*H*-pyrazolo[3,4-*b*]pyridine derivatives (**41**, **42**, **43**, **44**, and **45**, among others), have been designed as potential anticancer agents (Figure 12). They have been screened for their antitumor activity in vitro, with compound **41** having been tested with a full-panel, five-dose assay to assess its GI_50_, TGI, and IC_50_ values. Compound **41** exhibits broad-spectrum antiproliferative activities over the whole National Institute cancer (NCI) panel, with excellent growth inhibition. Its full-panel GI_50_ (the mean activity value for the entire panel, MG-MID) value equals 2.16 mM and subpanel GI_50_ (MG-MID) range is 1.92–2.86 mM. Moreover, 1*H*-pyrazolo[3,4-*b*]pyridines **41**, **42**, **43**, **44**, and **45** have been tested for their antiproliferative activity against a panel of leukemia cell lines (K562, MV4-11, CEM, RS4;11, ML-2, and KOPN-8), where they showed excellent antileukemic activity. These results make compounds **42**, **43**, **44**, and **45** encouraging lead molecules to stimulate optimization to frame more robust and efficient anticancer candidates [124].

Compounds **46** and **47** (Figure 13) demonstrated antitumor activity against a liver cell line with an IC_50_ of 3.73 µM and 3.43 µM, respectively [64].

The human nicotinamide phosphoribosyltransferase (NAMPT) is involved in the first step of the conversion of nicotinamide (NAM) to the biologically important enzyme co-factor nicotinamide adenine dinucleotide (NAD). Compound **48** (Figure 14) was identified after structure-based design studies and exhibits nanomolar antiproliferation activities against human tumor lines in in vitro cell culture experiments. This compound has an IC_50_ = 6.1 nM and an IC_50_ = 4.3 nM for NAMPT BC and A2780 cells, respectively [125].

The rat sarcoma virus (RAS) is a guanosine-nucleotide-binding protein. Specifically, it is a single subunit, small GTPase. Activated RAS GTPase signaling is a crucial motor of oncogenic alteration and malignant disease. The use of cellular models of RAS-dependent cancers has driven the identification of molecule **49** (SCH51344) (Figure 15) as a human mutT homolog MTH1 (also known as NUDT1) inhibitor, a nucleotide pool sanitizing enzyme [126].

Compound **50** (Figure 16) has been designed as a tubulin polymerization inhibitor targeting the colchicine site. It has been tested by MTT assays for its antiproliferative activity against three human cancer cell lines (SGC-7901, A549, and HeLa). **50** exhibits noticeable in vitro potential activity because SGC-7901 has an IC_50_ = 13 nM, and it could significantly inhibit tubulin polymerization and strongly disrupt the cytoskeleton [127].

An encouraging molecular target for non-small cells lung carcinoma (NSCLC) is the anaplastic lymphoma kinase (ALK). Compound **51** (Figure 17) has been described for its excellent inhibitory activity against ALK-L1196M (IC_50_ < 0.5 nM) and ALK-wt. Moreover, **51** shows a remarkable inhibition of ROS1 (IC_50_ < 0.5 nM) and displays excellent selectivity over c-Met. **51** resolutely suppresses proliferation of ALK-L1196M-Ba/F3 and H2228 cells boarding EML4-ALK via apoptosis and the ALK signaling blockade [128].

Compound **52** (Figure 18) has been found to have an excellent DYRK1B inhibitory enzymic activity with an IC_50_ = 3 nM, cell proliferation inhibitory activity (IC_50_ = 1.6 μM) against HCT116 colon cancer cells, and inhibitory activity in a patient-derived colon cancer organoids model and a 3D spheroids assay model of SW480 and SW620 [18,19].

Mitogen-activated protein kinase 4 (MKK4) has recently been identified as the main regulator in hepatocyte regeneration. A scaffold-hopping approach has been performed to obtain compounds **53** and **54** (Figure 19), which show high affinity to MKK4 in the low nanomolar range and a selectivity profile from a multiparameter-optimization due to the essential antitargets (MKK7 and JNK1) and off-targets (BRAF, MAP4K5, and ZAK) in the MKK4 pathway [28].

Pimitespib (**55**, TAS-116, Figure 20) shows an inhibitory effect on the phosphorylation of KIT-Asp818Tyr, which is a c-kit gene that contains a germline Asp820Tyr mutation at exon 17. This mutation causes multiple gastrointestinal stromal tumors (GISTs). This small molecule can decrease the cecal tumor volume in model mice; thus, it seems to inhibit in vivo tumor progression. These studies suggest that Pimitespib can be a potential drug to control multiple GISTs in patients with germline KIT-Asp820Tyr [129].

### 4.2. Anti-Inflammatory Agents

The use of 1*H*-pyrazolo[3,4-*b*]pyridines as scaffolds for anti-inflammatory agents corresponds to 13% of the overall biomedical applications of such structures. The main combination of substituents used for such biological activity is the disubstitution at R^3^ and R^5^ (7045 molecules, 36% of this group). A total of 1761 pyrazolo[3,4-*b*]pyridines (9% of this group) include a trisubstitution at positions R^1^, R^4^, and R^5^. Other substitution patterns used less often are the combinations R^3^-R^4^-R^6^, R^3^-R^4^, and R^1^-R^3^-R^4^-R^6^, with 981, 961, and 908 molecules, respectively. In this case, the combination of substituents in which R^1^ = R^4^ = R^6^ = H and R^3^ and R^5^ are heterocyclic rings covers 30% of the total number of structures claimed as having an anti-inflammatory activity (5856 compounds).

The repression of spleen tyrosine kinase (Syk) is an encouraging approach for the treatment of several allergic and autoimmune disorders, such as rheumatoid arthritis, asthma, and allergic rhinitis. Compound **56** (Figure 21) presents great Syk inhibitory activity (IC_50_ = 1.2 μM), representing a good lead compound for further optimization [130].

Although pulmonary hypertension (PH) is a cardiovascular disease, its known inflammatory basis led us to include it in this section [131]. Soluble guanylate cyclase (sGC) is an essential signal-transduction enzyme activated by nitric oxide (NO). The pathogenesis of cardiovascular and other diseases can be a consequence of impairments of the NO– sGC signaling pathway. The stimulation of sGC is a promising treatment for pulmonary hypertension (PH), a disease related to a poor prognosis. Pyrazolopyridines **57** (BAY 41-2272) and **58** (BAY 41-8543) (Figure 22) exhibited favorable effects in experimental models of PH, despite their being related to unfavorable drug metabolism and pharmacokinetic (DMPK) properties. Riociguat (**59**) (Figure 22) is a designed compound found by SAR exploration that improves the DMPK profile and shows excellent effects on pulmonary hemodynamics and exercise capacity in patients with PH. Riociguat was investigated in phase III clinical trials for the oral treatment of PH [132] and approved in the USA, Europe, and other regions for patients with pulmonary arterial hypertension and marketed by Bayer under the trade name Adempas.

Molecules **60** and **61** (Figure 23) exhibit promising anti-inflammatory activity against TNF-α and IL-6. In concrete, against IL-6 with 60–65% inhibition at 10 µM. Moreover, they show potent IL-6 inhibitory activity, with an IC_50_ of 0.2 and 0.3 µM, respectively [133].

The inhibition of p38a is one of the major targets in developing anti-inflammatory drugs due to its prominent role in regulating inflammatory cytokines, such as TNFα and IL-1. Molecule **62** (Figure 24) has potent p38a inhibitor activity and excellent in vivo activity upon oral administration in animal models of rheumatoid arthritis [134].

The study of the activity of molecule **63** (Figure 25) on macrophage growth, phagocytosis of FITC-zymosan, radical scavenging affinity against OH**^·^**, ROO**^·^**, and O_2_**^−^**, and macrophage binding affinity to fluorescein isothiocyanate-conjugated bacterial lipopolysaccharide (FITC-LPS), together with its affection for the inflammatory mediators (nitric oxide (NO), tumor necrosis factor-a (TNF-α), prostaglandin E-2 (PGE-2), cycloxygenase-2 (COX-2), and 5-lipoxygenase (5-LO)) in LPS-stimulated macrophages, have converted it into a promising multipotent anti-inflammatory agent [135].

4-Anilinopyrazolopyridine derivative **64** (Figure 26) led to the optimization of phosphodiesterase 4 (PDE4) inhibitors. Compounds **65** and **66** present improved therapeutic capacity with fewer side effects, being orally active compounds. Compound **66** shows much higher bioavailability than compounds **64** and **65** [136].

Compound **67** (Figure 27) was discovered after the optimization of a high-throughput screening hit. This molecule exhibits potent and selective inhibition activity against phosphodiesterase 4 (PDE4) with a pIC_50_ of 8.5, thus being a promising target for the treatment of chronic obstructive pulmonary disease (COPD). Moreover, molecule **67** also inhibits LPS-induced TNF-α production from isolated human peripheral blood mononuclear cells and has a promising rat pharmacokinetic profile for oral dosing [137]. Compound **68** (Figure 27) was found to have excellent potent inhibitory activity against PDE4. This compound was discovered after SAR studies of the 5-position. Thus, optimization using X-ray crystallography and computational modeling led to **68** with sub-nM inhibition of LPS-induced TNF-α production from isolated human peripheral blood mononuclear cells [138].

T cell activation and survival strongly depend on protein kinase C θ (PKCθ). Recent studies show that T cell responses associated with autoimmune diseases are PKCθ-dependent. Selective and potent inhibition of PKCθ is likely to block autoimmune T cell responses without compromising antiviral immunity. Molecule **69** (Figure 28) was discovered by using structure-based rational design and has shown to be a potent and selective PKCθ inhibitor [139].

### 4.3. Nervous System Agents

The use of pyrazolo[3,4-*b*]pyridines as a scaffold for the treatment of nervous system diseases corresponds to 10% of the overall biomedical applications. A total of 2799 molecules (20% of this category) presents a disubstitution R^3^ and R^5^, while 2044 compounds (14% of this group) include a trisubstitution at R^1^-R^4^-R^5^. Lower amounts of molecules (1337, 964, and 881 molecules) present substituents at R_1_-R_4_-R_5_-R_6_, R_3_-R_4_-R_6_, and R_1_-R_3_-R_4_-R_6_, respectively. In this category the situation concerning the nature of substituents present at the different positions is not so clear, but 1509 structures (10% of the total) present R^1^ = alkyl (or alky-substituted), R^3^ = H, R^5^ = carbonyl group, and R^6^ not equal to a hydrogen atom.

Adenosine is a neurotransmitter distributed through a wide variety of tissues in mammals. A_1_-adenosine receptor (A_1_AR) is an adenosine receptor that modulates adenosine effects together with A_2A_, A_2B_, and A_3_. Affinity data at A_1_AR, A_2A_AR, and A_3A_R in bovine membranes reveal that **70** (Figure 29) binds selectively and with a high-affinity A_1_AR over A_2A_AR and A_3_AR [140].

Compound **71** (LASSBio873, Figure 30) was structurally designed by using the analgesic and sedative drug Zolpidem as a lead compound, which is used to treat anxiety disorders linked to the neuronal inhibition induced by ɣ-aminobutyric acid (GABA), the main inhibitory neurotransmitter in the mammalian central nervous system (CNS). Molecule **76** presents not only a potent ability to induce sedation but also a potent central antinociceptive effect [141].

Corticotropin-releasing factor type 1 (CRF(1)) is a novel target for the treatment of depression, anxiety, and stress-related disorders. Pyrazole-based molecule **72** (Figure 31) potently binds the CRF(1) with a Ki = 2.9 nM and inhibits the adrenocorticotropic hormone (ACTH) release from rat pituitary cell culture with an IC_50_ of 6.8 nM, which is the key hormone governing stress response. This compound also shows a great ACTH reduction of 84–86% when it is given orally at 30 mg/kg doses [142].

Compound **73** (Figure 32), which displays its CNS action as a muscarinic M1 receptor agonist, is an efficient compound to diminish the locomotor activity in mice at a dose of 10 µmol/kg [143].

Pyrazoloquinoline **74** (Figure 33) has been described as a potent, selective, and orally active phosphodiesterase 10A inhibitor (PDE10A) for the potential treatment of schizophrenia. Compound **74** inhibits MK-801-induced hyperactivity at 3 mg/kg with an ED_50_ of 4 mg/kg and exhibits a 6-fold improvement between the ED_50_ for inhibition of MK-801-induced hyperactivity and hypolocomotion in rats [144].

The latest indicators demonstrate that of the two confirmed methods for measuring amyloid, the decrease in cerebrospinal fluid (CSF) amyloid β_1–42_ (Aβ_1–42_) may be an earlier sign of Alzheimer disease (AD) risk [145]. Compounds **75** and **76** (Figure 34) avoid the decrease in cell viability caused by Aβ_1–42_. **76** also prevents the upregulation of AChE induced by Aβ_1–42_. It also may act as an antagonist of voltage-sensitive calcium channels. Compound **77** exhibits potential for the treatment of Alzheimer disease [146].

To close this review, the search carried out in DrugBank (see our Introduction) has allowed us to establish that only two 1*H*-pyrazole[3,4-*b*]pyridines have already been approved: Riociguat **59,** commercialized by Bayer as Adempas (described above and approved in 2013) and Vericiguat **78** (Figure 35), a stimulator of soluble guanylate cyclase (sGC) approved by the FDA in January 2021 and commercialized by Merck as Verquvo for the treatment of systemic heart failure [147].

## 5. Conclusions

In this paper, we reviewed the substitution patterns of 1*H*-pyrazolo[3,4-*b*]pyridines (**1**), establishing the type of substituents mainly used at positions N1, C3, C4, C5, and C6. Such analysis has established that the 1*H*-isomers (substituted or unsubstituted at N1) predominate in a ratio of 3.6 to 1. Among 1*H*-pyrazolo[3,4-*b*]pyridines (**1**), two substitution patterns also are predominant: 3,4,6-trisubstitution and 3,5-disubstitution, both groups presenting mainly a hydrogen atom or methyl substituent at N1.

The complex landscape of the synthetic methods used for preparing such heterocycles has been analyzed and classified into two main methodologies: (a) formation of a pyridine ring into an existing pyrazole ring, and (b) the formation of the pyrazole ring into a preexisting pyridine ring. Most of the reactions found in the literature are of type (a) and use 3-aminopyrazole (substituted or not at N1) as the starting material, which generally acts as a 1,3-NCC-dinucleophile, reacting, therefore, with a 1,3-CCC-biselectrophile. The different subtypes depend on the nature of the 1,3-CCC-biselectrophile. Most of the type (b) constructions use hydrazine (or substituted hydrazine) onto a 2-chloro substituted pyridine ring bearing an aldehyde, ketone, ester, or cyano group at C3. Such groups determine the nature of the substituent present a C3 of the final 1*H*-pyrazolo[3,4-*b*]pyridine formed.

Finally, we have analyzed the potential biological uses of 1*H*-pyrazolo[3,4-*b*]pyridines **1**, establishing that such types of structures have been used for a wide range of biological targets. In fact, one-half of the molecules described were developed to discover biological activities. The main bioactivity indicators found include antitumor agents, anti-inflammatory agents, and nervous system agents that cover 36% of the overall biomedical applications. The most important biological targets and molecules developed have been summarized, showing the high versatility of these structures. Also, the substitution patterns and types of substituents used for each of the biological activities analyzed have been summarized.

The fact that, from the total number of references included in SciFinder since 1908, more than 50% correspond to the period from 2012 to 2022, showing an almost exponential increase, with half of them being patents, clearly indicates that this type of structure currently plays an important role as a scaffold for the development of drug candidates. The only drawbacks they present are the regioselectivity issues described in the synthetic section of this review and the fact that, in many cases, it is difficult to establish unequivocally the regioisomer formed based on spectroscopic techniques.

## Data Availability

Data sharing is not applicable.

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
