# Peer review of "1H-Pyrazolo[3,4-b]pyridines: Synthesis and Biomedical Applications"

_molecules, 2022, doi:10.3390/molecules27072237_

Round 1

Reviewer 1 Report

This article brings a review of the diversity structural analysis of pyrazolo[3,4-b]pyridines. As stated by the authors themselves, although there are already published reviews covering this heterocycle nucleus, the authors have decided to add a contribution focusing on synthesis and biological applications at the same time.  The authors use a systematic methodology to select the works covered by the review in light of the eligibility and exclusion criteria used (lines 45 – 60). Considering the relevance of the pyrazolo[3,4-b]pyridines nucleus, I think this review may be of interest to researchers in the field of heterocyclic and medicinal chemistry. However, it is noteworthy that this review is far from covering all pyrazolo[3,4-b]pyridine derivatives and their activities that have been reported in recent years. Also, here are some points to consider:

  1. To assure the reliability of the results, I suggest that the authors specify the period considered in the search and insert the list of all references found of structures 1 and 2 mentioned in the 2nd paragraph as supplementary material.
  2. Even though the authors state that they carried out a revision of the literature covering the structural, synthetic, and biological aspects of structures 1 (1H-) and 2 (2H-), they focused the analysis only on structure 1. Furthermore, despite the lower number of 2 compared to 1, the number of 2-substituted derivatives of 2 (19.000 – item 2.1) is far from irrelevant. Thus, it would be worth rewriting the 4th paragraph of the introduction section.
  3. Since nitrogen is substituted (R1), compound 6 (item 3.1) must be presented as a 5-aminopyrazole.
  4. The authors did not cite references to scheme 2.
  5. At item 3.1.1. it would be important to report that several authors correlate the regioisomer obtained with the reaction conditions (acid or not acid). Furthermore, there are several examples in the literature of pyrazolo[3,4-b]pyridines substituted at the R4-position which were obtained from the reaction of 5-amino pyrazole and 1,3-dicarbonyl compounds. Such examples can easily be found in a more comprehensive search.
  6. As addressed in the text, there is no consensus on which of the two nucleophiles reacts in the first place. Therefore, it is not worthwhile to provide only one mechanism hypothesis for the formation of structure 1.
  7. There has been a misinterpretation on page 25, once the inflammatory process in Chagas disease is not the focus of the presented molecules. Considering the data showed (antiparasitic activity), these compounds should be placed as antiparasitic or anti-infective agents. In addition, there are also several examples of derivatives that have been reported as anti-infective agents for other infective diseases. This misinterpretation alters the number of molecules considered by the authors as anti-inflammatory agents (item 4 and Figure 7).
  8. There are two slices of “Antidiabetic agents” in the pie chart (Figure 7), which leads to a misinterpretation of the data presented. I strongly recommend the review of all the data in the graph.
  9. Considering that it isn’t appropriate to show new data in the conclusion section, I suggest placing the data and method used for the search carried out at Drugbank (lines 832 -837) in another section.
  10. Other specific comments:
    • Figure 3 summarizes the data from Tables 1-5. Also, the authors could insert the percentage in the pie chart. So, I think there is no point in keeping tables 1-5, and I suggest deleting them.
    • Replace “Heterocy” by Heterocycle in Figures 3 – 5, as well as “refl” and “ref” by reflux in the schemes. Also, avoid the abbreviation “rt” in the text and use the correct degree symbol (° C) for degree Celsius.
    • In Figure 19, compound 53 doesn’t have one sulfur atom marked in yellow, as established by the authors as standard for the structures.

Reviewer 2 Report

In this contribution, Borrell and coworkers discuss synthesis and biomedical applications of pyrazolo[3,4-b]pyridines. The overall subject discussed in the manuscript is interesting and therefore a fitting contribution to Molecules. However, the manuscript needs cleaning up to make it more understandable to the general community. The authors may wish to consider the comments below.

  1. Several reviews have been published on this topic before, covering different aspects of pyrazolo[3,4-b]pyridines. Therefore, authors must include an introduction in which the state of the art must be covered, highlighting the novelties of this review with respect to other published reviews.
  2. Decimal after digits is confusing, check and correct it.
  3. There are various Figures in the manuscript which can be merged.
  4. Section 4: Authors are suggested to include a sub-section on structure property relationship of the discussed compounds.
  5. Authors are also suggested to include a section on Future Perspectives and Suggestions before conclusion.

Reviewer 3 Report

The manuscript "Pyrazolo[3,4-b]pyridines: Synthesis and Biomedical Applications" is devoted to an interesting and important topic. The review contains an interesting analysis of possible substituents in pyrazolo[3,4-b]pyridines. However, the review contains several undesirable moments. In fact, this review is a compilation of 3 different reviews which are not connected to each other. They are: analysis of substituents, review on synthesis and review on biological activity. Analysis on substituents seems to be original and novel. Review on synthesis looks like incomplete doubling on the reviews from ref. 5-8. Review on biological activity is just a narration of papers on three types of biological activities of pyrazolo[3,4-b]pyridines, and no analysis of SAR or something like this is provided. These three parts are not connected to each other. There is no analysis inside of them. They represent common narration.

In my opinion, the manuscript should be rejected or totally revised. In case of revision, the next issues should be considered:

  1. Why only scifinder was used for search? Reaxys and scifinder should be used both, since there are many paper that are covered by one of them and are not covered by the other (especially old papers).
  2. Time period of the review on synthesis should be changed. Now it is all the time, but it should be started from the papers that were not covered by previous reviews (ref. 5-8). Or a strong explanation on the necessity of repeated revision of the syntheses for all the time. During reading of reviews from ref. 5-8, I noted that there are some approaches to pyrazolo[3,4-b]pyridines that are different from two described in the present manuscript. So, I strongly recommend to change the time period for the review and expand the types of the approaches.
  3. Provide SAR analysis in the part with biological activity.

Round 2

Reviewer 3 Report

The authors have improved the manuscript. Now it is ready to be accepted.